# Comparison of Aneurysmal and Non-Aneurysmal Spontaneous Cervical Artery Dissections in a Large Multicenter Cohort

**Valentin K. Steinsiepe** [1], **Hakan Sarikaya** [1,2], **Pasquale R. Mordasini** [3,4], **Susanne Wegener** [2], **Corinne Inauen** [2], **Philipp Baumgartner** [2], **Simon Jung** [1], **Kateryna Antonenko** [1], **Urs Fischer** [1], **Jan Gralla** [4], **Roza M. Umarova** [1], **Barbara Goeggel Simonetti** [1], **Constance J. H. C. M. van Laarhoven** [5], **Gert J. de Borst** [5], **Hugues Chabriat** [6], **Mirjam R. Heldner** [1,†] and **Marcel Arnold** [1,*,†]

1 Department of Neurology, University Hospital of Bern, 3010 Bern, Switzerland; valentin.steinsiepe@insel.ch (V.K.S.)
2 Department of Neurology, University Hospital of Zurich, 8091 Zürich, Switzerland
3 Network Radiology, Cantonal Hospital of St. Gallen, 9000 St. Gallen, Switzerland
4 Department of Neuroradiology, University Hospital of Bern, 3010 Bern, Switzerland
5 Department of Vascular Surgery, University Medical Center of Utrecht, Utrecht University, 3584 Utrecht, The Netherlands
6 Translational Neurovascular Centre and Department of Neurology, Hôpital Lariboisière, FHU NeuroVasc, Université Paris Cité Paris and INSERM U1141, 75010 Paris, France
* Correspondence: marcel.arnold@insel.ch; Tel.: +41-31-632-70-00
† These authors contributed equally to this work.

**Abstract:** Dissecting aneurysms in patients with spontaneous cervical artery dissections have, so far, been reported as "benign", but more specific information is scarce. We aimed to elucidate (1) vascular risk factors, (2) local and ischemic symptoms, and (3) long-term prognosis compared to non-aneurysmal dissections. This case–control study included consecutive patients with spontaneous cervical artery dissection from three university hospitals in Switzerland and France, evaluated at baseline and at 3 months. In addition, further follow-ups were performed at the discretion of the treating physician. Dissecting aneurysms were diagnosed with duplex sonography, magnetic resonance angiography, and/or digital subtraction angiography. Of 1012 patients, 151 (14.9%) presented with 167 dissecting aneurysms at baseline ($n = 103$) or follow-up ($n = 64$). The median follow-up was 24.9 months (IQR: 6.8–60.8). Compared to patients without a dissecting aneurysm there were no significant differences in the vascular risk factors or local symptoms (91.4 vs. 89.8%). Ischemic strokes at baseline were less common (29.1% vs. 54.4%; OR: 0.41; 95% CI: 0.28–0.60) in patients with a dissecting aneurysm, even after correction for the degree of stenosis of the dissected arteries (OR: 0.53; 95% CI: 0.34–0.81). Patients with a dissecting aneurysm more often had a favorable clinical outcome (modified Rankin Scale Score of 0–1) at 3 months (80.6% vs. 54.5%). There was no significant difference in recurrent cerebrovascular events at 3 months or overall. The lower rate of ischemic strokes at baseline may reflect a different pathogenic mechanism, such as a smaller initial tear in the vessel wall or an increased vessel caliber from an early or primary intramural hematoma with a different shape.

**Keywords:** cervical artery dissection; dissecting aneurysm; vertebral artery; internal carotid artery

## 1. Introduction

Spontaneous dissections of the cervical arteries (SCADs) occur when a hematoma within the tunica media detaches from the layers of the extracranial carotid or vertebral artery without a preceding major trauma.

The hematoma is thought to form from an intimal tear or rupture of the vasa vasorum [1]. Subsequent inward or outward protuberance of the false lumen results in a stenosis, occlusion, and/or dissecting aneurysm (DA) [2]. DAs have been observed in

13.6–39% of patients with spontaneous internal carotid artery dissection (ICAD) [3–7] and in 8.3–23% of patients with vertebral artery dissection (VAD) [5–7]. They usually develop and get diagnosed within one month of dissection, but some have been discovered after 3–9 months [3,4,7,8]. DAs in SCADs have been described to follow a relatively benign course with few complications, especially with few ischemic events and a low tendency to increase in size [3–5,9,10]. In traumatic cases, however, it has been reported that they seem more likely to grow and tear [5,11,12].

Apart from these characteristics the natural history of DAs in SCADs is poorly defined. Most published data originate from retrospective cohorts of fewer than 50 patients, and many studies are purely descriptive or lack long-term follow-up. Analyses do not always differentiate among extracranial, intracranial, vertebral, and carotid, or between spontaneous and traumatic dissections. Direct comparisons between patients with and without DAs are rare [7,8,10].

In this study, we aim to elaborate the differences and similarities between aneurysmal and non-aneurysmal SCADs. We hypothesized that the aneurysmal subtype is more likely to cause local symptoms through compressive mechanisms and carries a similar long-term prognosis.

## 2. Materials and Methods

This case–control study is a retrospective analysis of prospectively collected data. Consecutive patients with SCADs at the University Hospital Bern, Switzerland (10/1990–01/2013), the University Hospital Zurich, Switzerland (10/1990–07/2011), and the University Hospital Lariboisière, Paris, France (01/1997–01/2013), who presented to their respective emergency departments were included and then followed up to prospectively assess their outcomes. All three centers are part of the Cervical Artery Dissection and Ischemic Stroke Patients (CADISP Plus) consortium. Data collection protocols were published previously [6]. For the retrospective analysis, the data were grouped based on whether the patient had any radiological evidence of a DA on any vessel at baseline or at any follow-up. SCADs without any information on DA status were excluded.

The inclusion criteria were a first-time SCAD of the internal carotid artery or vertebral artery in patients aged ≥16 years. The exclusion criteria were major trauma, a previous SCAD, or intracranial extension of a SCAD.

The baseline evaluation of all patients included a structured patient interview, screening for vascular risk factors, clinical examination, and assessment for local symptoms and ischemic events by a board-certified neurologist; routine blood sampling, electrocardiography, and duplex sonography assessed by two board-certified neurologists; cerebral and cervical magnetic resonance imaging and angiography (MRA), computed tomography angiography (CTA), and/or digital subtraction angiography (DSA) assessed by a board-certified neuroradiologist and a neurologist; and a reconstruction of the data from hospital reports [13].

Follow-up was carried out clinically or by a structured telephone interview at 3 months and then at the discretion of the treating physician, usually after 6, 12, and 24 months. The predefined outcome variables included early (within 3 months) and late recurrent events (SCAD, transient ischemic attack (TIA), ischemic stroke), clinical functional outcome using the modified Rankin Scale (mRS), and death at 3 months. An mRS score of 0–1 was defined as a favorable clinical outcome at 3 months. The mRS was only evaluated in stroke patients.

The diagnoses of SCADs were made by MRA, CTA, and/or DSA showing a mural hematoma, double lumen, string sign, intimal valves, or DA of the internal carotid artery or vertebral artery. DAs were identified using duplex sonography, MRA, and/or DSA and defined as a fusiform dilatation or saccular aneurysm that was not visible on the contralateral vessel [13].

Arterial hypertension was defined as blood pressure >160/95 mmHg before the year 2000 and >140/90 mmHg afterward or by antihypertensive treatment before a SCAD. Hypercholesterolemia was defined as fasting cholesterol >5.0 mmol/L or by lipid-lowering

treatment before a SCAD. Current smoking was recorded in the last 5 years. Connective tissue disorders studied included Ehlers–Danlos syndrome type IV, Marfan syndrome, and osteogenesis imperfecta type I.

Local symptoms were defined as any headache or cervical pain, tinnitus, Horner's syndrome in the case of carotid dissection, cranial nerve palsy, or radiculopathy. Ischemic symptoms were defined as ischemic stroke, TIA, amaurosis fugax, retinal infarction, and spinal infarction.

The data were collected using pre-printed forms and then transferred to a digital format. Data processing and analysis were performed using R version 3.6.2, including the packages tidyverse 1.3.0, readxl 1.3.1, lubridate 1.7.8, mice 3.10.0.1, glmnet 4.0–2, and survival 3.1–12, and SPSS 25.0 (SPSS Inc., Chicago, IL, USA).

Statistical methods to compare the baseline characteristics, demographic data, vascular risk factors, imaging findings, and therapy details between the aneurysmal and non-aneurysmal SCAD patients included two sample unpaired *t*-tests with equal variances, $\chi^2$ tests, and Mann–Whitney U tests where appropriate. $\alpha$ was set to 5%. The reported *p*-values were corrected for multiple testing by the Benjamini–Hochberg method.

Variable selection was performed with lasso regression (the lambda was set to the minimum mean cross-validated error within one standard error of minimum value). The variables "local symptoms" and "ischemic symptoms" were excluded from the variable selection to avoid multicollinearity. Variables concerning only a subset of the data were excluded from variable selection.

Distributions of the numerical variables were assessed by q-q-plots, and range checks were performed. Variables with more than 50% missing values (mural hematoma on imaging, fibromuscular dysplasia on DSA, weight, height), outliers, and implausible values were removed. Several categorical variables were dichotomized.

We performed a sensitivity analysis by dividing the patients by aneurysm status at baseline only instead of at baseline and follow-up. There were no relevant changes in our results. The analysis is shown in the Appendix A (Table A1).

Of the original 1230 patients, we excluded 73 due to failing our prespecified criteria (Figure 1). A further 145 patients did not have their aneurysm status recorded, leaving us with a final set of 1012 patients. The variables of fibromuscular dysplasia and BMI contained >50% missing variables and were, therefore, discarded.

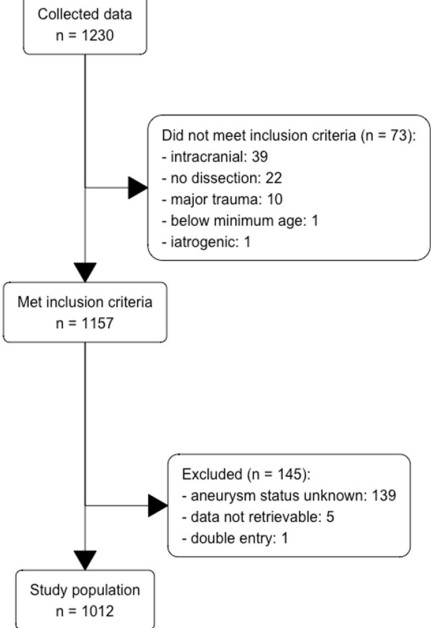

**Figure 1.** Flow diagram for inclusion and exclusion of patients with SCADs.

## 3. Results

### 3.1. Description of Patients

Our data consisted of 1012 patients with 1231 SCADs. The patients had a median age of 45 years, and 44.6% were women. In 63.0%, the dissection was located on the ICA, in 30.2% on the VA, and in 6.7% on both. A total of 17.8% had multiple dissections. At baseline, 94.8% of the patients had a duplex sonography, 76% had a MRI/MRA, 67.1% had a CT/CTA, and 24.4% had a DSA. The median length of follow-up was 24.9 months (IQR: 6.8–60.8 months).

Overall, the most common vascular risk factors were hypercholesterolemia (53.2%), smoking (27.7%), minor trauma (22.6%), and high blood pressure (22.4%). The most common clinical features at baseline were headache (71.8%), cervical pain (45.9%), ischemic stroke (50.6%), and Horner's syndrome (36.1%). A total of 512 patients had an ischemic stroke. In the subset of 403 patients with an ischemic stroke and complete data collection, 303 (75.2%) had an mRS of 0–2 after 3 months, whereas 91 (22.6%) had an mRS of 3–5. A total of 9 patients (2.2%) died.

Recurrent SCADs occurred in 4.4% of the patients. A total of 22 patients (2.2%) had a recurrent ischemic stroke, 13 (1.4%, median of 24.5 days) of which occurred in the first 3 months and another 8 (0.8%, median of 3.3 years) thereafter. The date of the recurrent ischemic stroke of one patient with a DA was missing. A total of 34 patients (3.6%) had a recurrent TIA, with 18 before and 16 after 3 months.

### 3.2. Comparison of Patients with and without DAs

A total of 151 patients (14.9%) presented with 167 DAs at baseline or follow-up. A total of 103 DAs (61.7%) were diagnosed at baseline. We also estimated the hazard of diagnosing a new DA, which seemed to be negligible after 3 months (Figure 2).

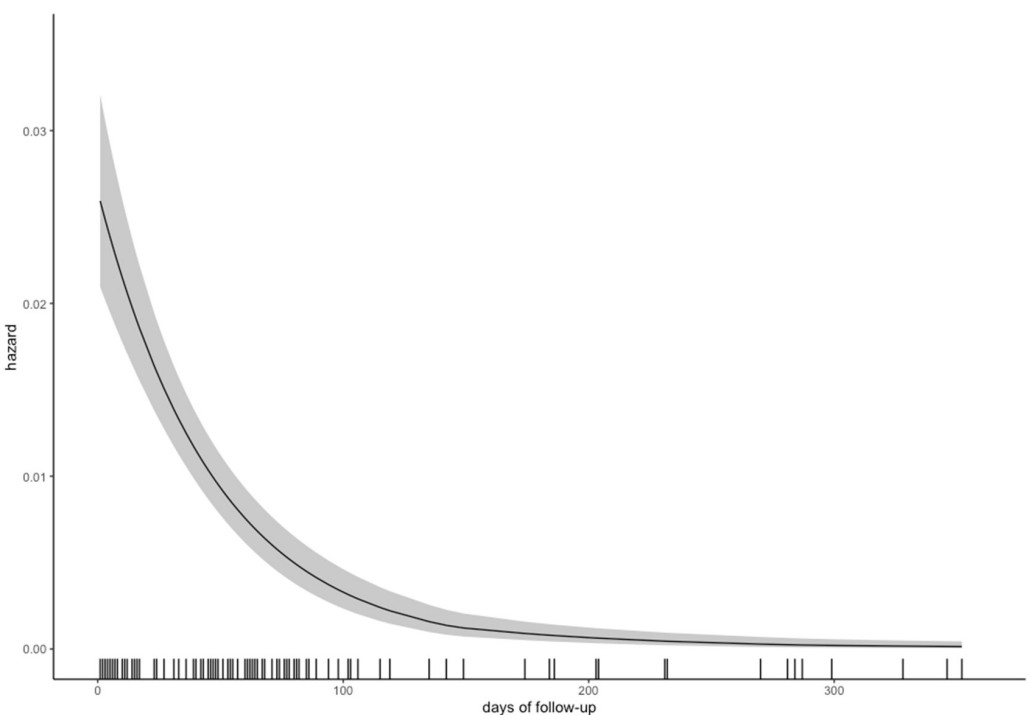

**Figure 2.** Hazard of diagnosing a new dissecting aneurysm over time, with the 95% confidence interval in grey.

The overall completeness of follow-up was comparable among the groups (41.3% in patients with DAs vs. 45.1% in patients without), reflecting little risk for informative censoring [14]. Figure 3 shows a Lexis diagram for all patients divided by DA status.

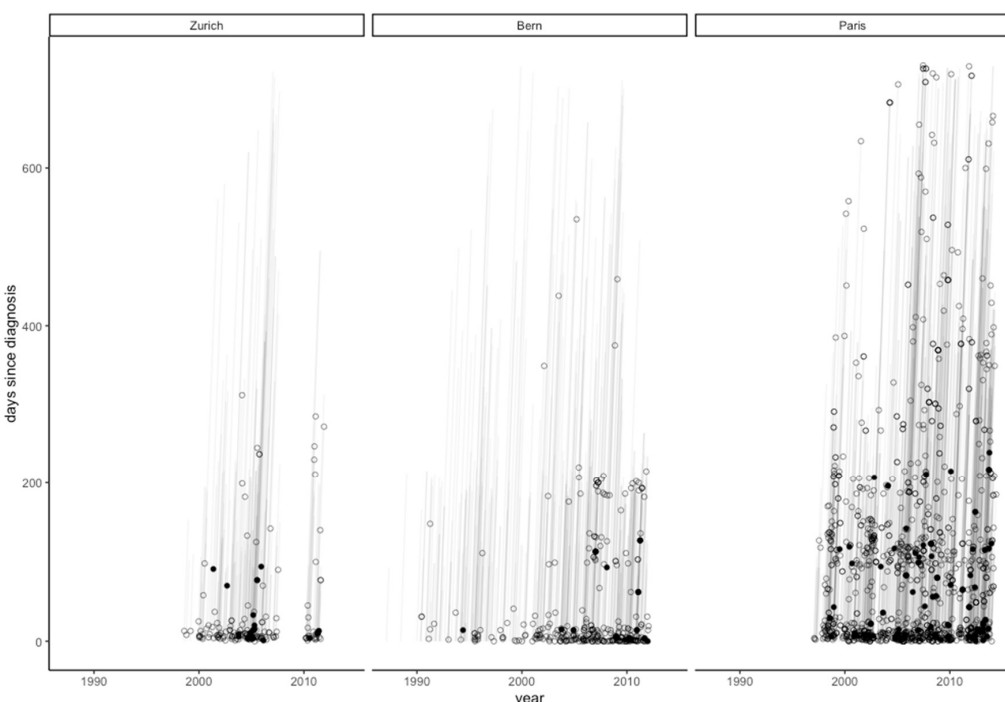

**Figure 3.** Lexis diagram of MRAs performed on SCAD patients by each university hospital during the first two years of follow-up. Every circle signifies an MRA, and every filled circle signifies the first diagnosis of a dissecting aneurysm.

DAs occurred mostly in patients with single SCADs, but were more common in patients with multiple SCADs than non-aneurysmal dissections. Only 15 patients (10%) presented with more than one DA. Patients without DAs tended to have higher grades of stenosis and a higher rate of occlusion (38.8 vs. 16.7%) (Table 1).

**Table 1.** Distribution of 1012 patients among centers and characteristics of dissections by aneurysm status.

| | Number of Patients Included in Analysis *n* (% Missing Data) | No Dissecting Aneurysm *n* (%) | Dissecting Aneurysm *n* (%) |
|---|---|---|---|
| Hospital | 1012 (0.0) | | |
| Zurich | | 131 (15.2) | 17 (11.3) |
| Bern | | 315 (36.6) | 21 (13.9) |
| Paris | | 415 (48.2) | 113 (74.8) |
| Number of dissections | 1012 (0.0) | | |
| 1 | | 728 (84.6) | 105 (69.5) |
| 2 | | 116 (13.5) | 31 (20.5) |
| 3 | | 15 (1.7) | 9 (6.0) |
| 4 | | 2 (0.2) | 6 (4.0) |
| Number of aneurysms | 1012 (0.0) | | |
| 1 | | 0 (0.0) | 136 (90.1) |
| 2 | | 0 (0.0) | 14 (9.3) |
| 3 | | 0 (0.0) | 1 (0.7) |
| Degree of stenosis | 858 (15.2) | | |
| None | | 58 (8.0) | 33 (25.0) |
| <50% | | 111 (15.3) | 23 (17.4) |
| 50–80% | | 46 (6.3) | 15 (11.4) |
| 80–99% | | 229 (31.5) | 39 (29.5) |
| Occlusion | | 282 (38.8) | 22 (16.7) |

Tables 2 and 3 show comparisons of the patients with and without a DA. In the univariate analysis, the patients with a DA were significantly less likely to present with an ischemic stroke at baseline (29.1 vs. 54.4%), but did not differ in local symptoms compared to the patients without a DA (91.4 vs. 89.8%). All other baseline variables were similar in the compared groups.

**Table 2.** Comparison of the characteristics of 1012 patients by aneurysm status; univariate analysis.

| | Number of Patients Included in Analysis | No Dissecting Aneurysm | Dissecting Aneurysm | Univariate Analysis |
|---|---|---|---|---|
| | *n* (% Missing Data) | Median (IQR)/ *n* (%) | Median (IQR)/ *n* (%) | *p*-Value |
| Age | 1012 (0.0) | 45.0 (38.0 to 52.0) | 46.0 (38.5 to 52.0) | 0.77 |
| Female sex | 1012 (0.0) | 373 (43.3) | 78 (51.7) | 0.368 |
| Days to diagnosis | 1012 (0.0) | 5.0 (2.0 to 12.0) | 7.0 (3.0 to 14.5) | 0.143 |
| Vessel affected | 1012 (0.0) | | | 0.701 * |
| ICA | | 551 (64.0) | 87 (57.6) | |
| VA | | 268 (31.1) | 38 (25.2) | |
| ICA + VA | | 42 (4.9) | 26 (17.2) | |
| Multiple dissections | 1012 (0) | 134 (15.6) | 46 (30.5) | 0.441 * |
| Fever and infection | 927 (8.4) | 85 (10.8) | 9 (6.3) | 0.417 |
| Minor trauma | 1009 (0.3) | 190 (22.1) | 38 (25.5) | 0.626 |
| Connective tissue disease | 1006 (0.6) | | | 0.484 |
| Family history | | 15 (1.8) | 2 (1.3) | |
| Diagnosed | | 15 (1.8) | 6 (4.0) | |
| Family history of ischemic stroke | 994 (1.8) | 121 (14.3) | 16 (10.7) | 0.569 |
| Smoking | 1003 (0.9) | | | 0.731 |
| Former | | 117 (13.7) | 24 (16.1) | |
| Current | | 242 (28.3) | 38 (25.5) | |
| Arterial hypertension | 1011 (0.1) | 188 (21.9) | 38 (25.2) | 0.626 |
| Hypercholesterolemia | 849 (16.1) | 390 (53.8) | 62 (50.0) | 0.668 |
| Migraine | 979 (3.3) | | | 0.668 |
| Without aura | | 159 (19.1) | 30 (20.3) | |
| With aura | | 82 (9.9) | 19 (12.8) | |
| Diabetes mellitus | 1011 (0.1) | 18 (2.1) | 4 (2.6) | 0.701 |
| Hormonal anticonception | 435 (3.5) | 115 (31.9) | 16 (21.3) | 0.387 |
| Local symptoms | 1012 (0.0) | 773 (89.8) | 138 (91.4) | 0.731 |
| Ischemic symptoms | 1012 (0.0) | 621 (72.1) | 70 (46.4) | <0.0001 |
| Headache | 1010 (0.2) | 614 (71.5) | 111 (73.5) | 0.751 |
| VAS headache | 447 (38.3) | 7.0 (5.0 to 8.0) | 7.0 (6.0 to 9.0) | 0.626 |
| Thunderclap headache | 900 (11.1) | 48 (6.3) | 13 (9.8) | 0.417 |
| Cervical pain | 1009 (0.3) | 391 (45.5) | 72 (48.0) | 0.731 |
| VAS cervical pain | 249 (46.2) | 6.0 (5.0 to 8.0) | 6.5 (5.0 to 8.0) | 0.491 |
| Radiculopathy | 1004 (0.8) | 4 (0.5) | 1 (0.7) | 0.701 |
| Spinal infarction | 1012 (0.0) | 1 (0.1) | 0 (0.0) | 1 |
| Subarachnoid hemorrhage | 1010 (0.2) | 13 (1.5) | 7 (4.6) | 0.143 |
| Horner's syndrome in carotid dissection | 706 (0.0) | 280 (47.2) | 54 (47.8) | 1 |
| Tinnitus | 1010 (0.2) | 63 (7.3) | 18 (11.9) | 0.373 |
| Cranial nerve palsy | 1012 (0.0) | 66 (7.7) | 15 (9.9) | 0.626 |
| Amaurosis fugax | 1010 (0.2) | 64 (7.5) | 7 (4.6) | 0.565 |
| Retinal infarction | 1012 (0.0) | 5 (0.6) | 2 (1.3) | 0.565 |
| Transient ischemic attack | 1010 (0.2) | 250 (29.1) | 34 (22.7) | 0.417 |
| Number of transient ischemic attacks | 275 (3.2) | 1.0 (1.0 to 2.0) | 1.0 (1.0 to 1.0) | 0.626 |
| Ischemic stroke | 1012 (0.0) | 468 (54.4) | 44 (29.1) | <0.0001 |
| NIHSS baseline | 503 (1.8) | 5.0 (2.0 to 14.0) | 4.0 (1.0 to 11.0) | 0.417 |

**Table 2.** *Cont.*

| | Number of Patients Included in Analysis | No Dissecting Aneurysm | Dissecting Aneurysm | Univariate Analysis |
|---|---|---|---|---|
| | *n* (% Missing Data) | Median (IQR)/ *n* (%) | Median (IQR)/ *n* (%) | *p*-Value |
| Early recurrent stroke | 954 (5.7) | 8 (1.0) | 5 (3.4) | 0.209 |
| Late recurrent stroke | 954 (5.7) | 6 (0.7) | 2 (1.4) | 0.484 |
| Early recurrent TIA | 955 (5.6) | 13 (1.6) | 5 (3.4) | 0.46 |
| Late recurrent TIA | 955 (5.6) | 16 (2.0) | 0 (0.0) | 1 |
| Recurrent dissection | 957 (5.4) | 35 (4.3) | 7 (4.8) | 0.626 |
| Rankin Scale after 3 months | 403 (21.3) | 1.0 (1.0 to 3.0) | 0.0 (0.0 to 1.0) | 0.007 |
| Favorable Rankin Scale after 3 months | 403 (21.3) | 200 (54.5) | 29 (80.6) | 0.048 |
| death after 3 months | 825 (18.5) | 9 (1.3) | 0 (0.0) | 0.633 |

\* *p*-value calculated per dissection, not per patient; ICA: internal carotid artery; VA: vertebral artery; IQR: inter-quartile range.

**Table 3.** Comparison of the characteristics of 1012 patients by aneurysm status; multivariate analysis.

| | No Dissecting Aneurysm | Dissecting Aneurysm | Multivariate Analysis | |
|---|---|---|---|---|
| | *n* (%) | *n* (%) | Odds Ratio (95% CI) | *p*-Value |
| Ischemic stroke | 468 (54.4) | 44 (29.1) | 0.41 (0.28–0.60) | <0.0001 |

CI: confidence interval.

We did not find a significantly higher rate of recurrent ischemic strokes in the first three months or thereafter (3.4 vs. 1.0% and 1.4 vs. 0.7%, respectively), early or late recurrent TIAs (3.4 vs. 1.6% and 0.0 vs. 2.0%, respectively) or recurrent dissections (4.8 vs. 4.3%) for patients with DAs. The mRS after 3 months was significantly better (0 vs. 1 point) and a favorable mRS after 3 months was significantly more likely (80.6 vs. 54.5%). Death after 3 months did not differ significantly (0 vs. 1.3%).

Multivariable analysis corrected for stenosis (complete case analysis, *n* = 858) showed an independent association of DAs with ischemic stroke at baseline (OR: 0.53, 95% CI: 0.34–0.81).

## 4. Discussion

### 4.1. Risk Factors

We were unable to demonstrate a significant association between vascular risk factors and DAs in our data, which is in accordance with the existing literature. Arterial redundancies and multiple dissections have been associated with DA formation, but the latter was calculated per patient rather than per vessel; the likelihood of developing at least one DA is naturally higher with more than one dissected vessel [10]. Other authors were not able to highlight a specific vascular risk factor for DAs [7].

Of note, the rates for current smoking, migraine, and minor trauma were very similar in both groups. It is, therefore, possible that the formation of a DA or stenosis depends less on structural abnormalities of the vessel wall and its surroundings and mainly on purely mechanical factors, such as the position of the original tear between the tunica intima and the tunica adventitia and the size of the defect. In animal models, larger tears were shown to lead to stenotic lesions, smaller tears were shown to cause DAs, and even smaller tears were shown to heal spontaneously [15].

### 4.2. Symptoms

We did not find that DAs were more often associated with local symptoms, conversely to what was expected. Local symptoms, especially lower cranial nerves palsies, are traditionally explained by the compressive effect of an enlarged carotid artery, and therefore,

thought to be associated with DAs [16]. In addition, hypoperfusion due to an impaired or compressed vasa nervorum has been discussed as a potential cause of cranial nerve palsies [17]. In the present study, we found similar rates for Horner's syndrome, cranial nerve palsies, and tinnitus when comparing the groups. Furthermore, our data showed no difference between the rates or VAS scores of headache and neck pain.

Patients with DAs had significantly fewer ischemic strokes at baseline in our analysis. Although most strokes in CAD are due to embolism [18], we initially hypothesized that this finding was related to the degree of stenosis [19], postulating that eccentric protrusion of the dissection in a DA causes less high-grade stenosis and occlusion, which would lead to less flow turbulence or restriction, and therefore, to fewer embolisms or reduced perfusion, respectively. However, the lower rate of ischemic stroke after correction for the degree of stenosis in our secondary analysis (Figure 4) contradicts this reasoning. One could postulate that we are dealing with two facets of the same pathophysiological process and that mechanical factors, namely the characteristics and the size of the initial tear, the shape of the intramural hematoma, an increased vessel caliber, or different pathomechanisms, such as the primary rupture of the vasa vasorum, may be the main difference between aneurysmal and non-aneurysmal SCADs, leading to the lower rate of ischemic strokes we have seen in our data [17].

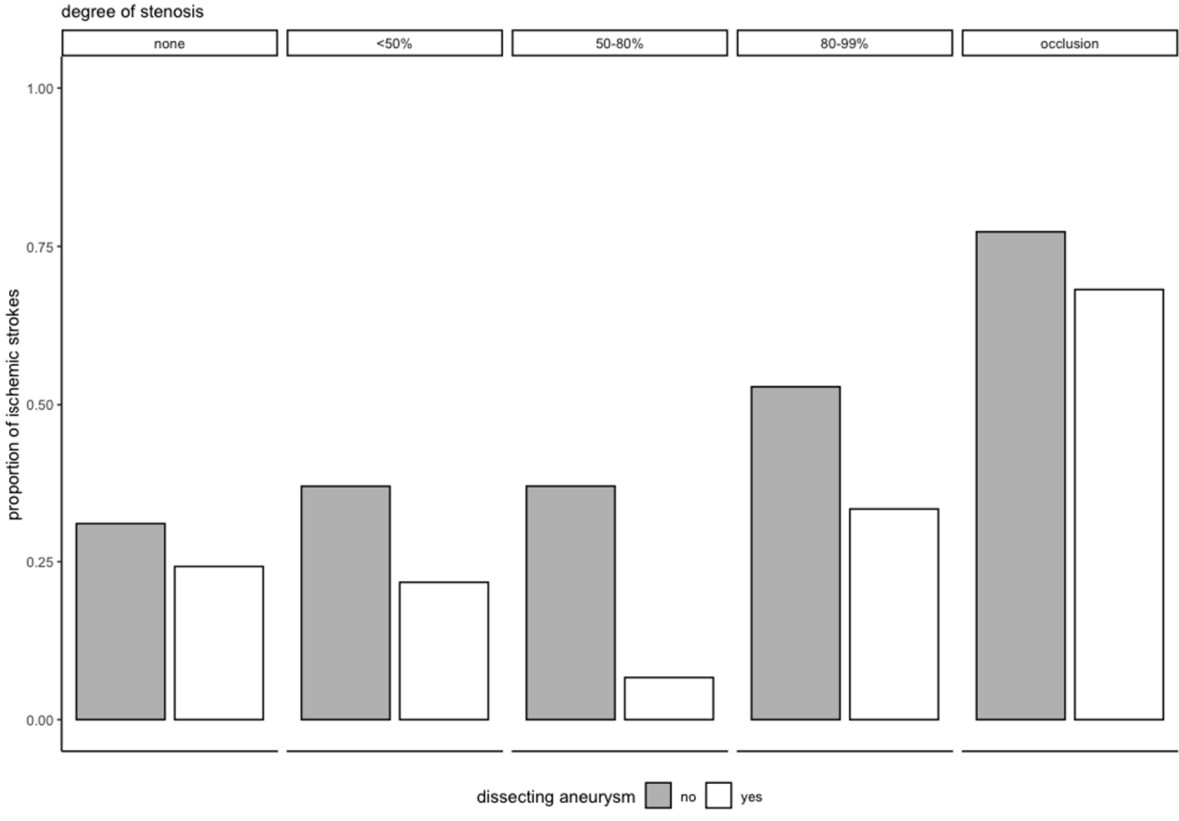

**Figure 4.** Proportion of ischemic stroke at baseline by degree of stenosis and aneurysm status.

### 4.3. Outcome

We were not only unable to show a difference in outcome but also unable to show a difference in recurrent ischemic strokes between patients with and without DAs. However, due to the low number of events, our analysis is underpowered and does not allow for a definitive conclusion. There is also a methodological issue: In most studies, DAs were often diagnosed at two timepoints (usually at baseline and after 3 months, sometimes up to 9 months) and then followed up [3,9,10]. This creates uncertainty whether a recurrent stroke within the diagnostic window occurred before or after aneurysm formation. To make our work comparable to previous studies, we divided the early and late recurrent

ischemic strokes. This seems sensible from a pathophysiological standpoint as well; reports of patients with ischemic strokes from a persistent DA do exist but have to be distinguished from ischemic strokes in the acute setting of SCADs [11,20].

Our results fit well into the existing literature: late ischemic strokes have been noted in about 1% (0–2.6%) of patients with DAs in studies that have specified the time of the ischemic stroke, and therefore, have to be considered rare [3,4,7,9,10,21]. Similarly, microemboli on transcranial doppler sonography were not associated with aneurysmal SCADs [18].

*4.4. Time Course*

Usually a significant proportion of DAs has only been discovered during follow-up imaging within the first months, and while this number varies greatly, it seems to be lower in cohorts that have mainly used conventional or digital subtraction angiography (0–20.8%) [4,19,21] than in cohorts that used mostly MRA (14.3–62.5%) [3,7,9,10]. Some DAs may, therefore, change size, but others may simply be missed in the first imaging. One reason may be that in early MRA-TOF, the DA pouch can be mistaken for an acute wall hematoma [4,22]. Additionally, due to delayed filling in stenotic vessels, the DA may not be visualized at all. Since most stenoses gradually diminish over the first months [23], "new" DAs might be found.

In our study, the hazard of diagnosing a new DA was the highest during the first 3 months (Figure 2), confirming the need for the abovementioned diagnostic window; most DAs (103, 61.7%) were diagnosed at baseline and only two after one year. This is consistent with a recent study (including 88 Bernese patients shared by our data), which showed that 74% of DAs were diagnosed at baseline, and the remaining ones with a median time delay of 6.2 months [8]. It is also in line with the data from serial MRIs, where changes of the intramural hematoma occurred up to 6 months after the event, but where the majority of hematomas had already resolved after 3 months [24]. Because of the intervals between the MRAs, the hazard in this paper was calculated using the midpoint of that interval—that is, we assumed that the DA occurred between the diagnostic and the previous imaging. Due to the delayed development of aneurysms, one could argue that patients should be followed more closely during the first few months or after the development of a new aneurysm. However, our findings and the literature do not support this approach, since extracranial DAs seem to carry minor risk of recurrent clinical events.

**5. Limitations**

The main limitation of this study is its retrospective design. Additionally, the data presented do not discern between aneurysms at baseline and aneurysms formed during follow-up. We did, however, perform a sensitivity analysis, which showed no significant change in the results (Tables A1 and A2). In addition, data on the size and shape of the DAs were missing. Our analysis was also insufficient for finding differences in recurrent ischemic strokes, and the mid-term follow-up data were assessed at different time points.

**6. Conclusions**

In our study, we could not identify vascular risk factors or local symptoms that distinguished aneurysmal from non-aneurysmal SCADs. The lower risk of ischemic strokes at baseline, which was independent of the degree of stenosis in our data, is a novel finding and may reflect a different pathomechanism in aneurysmal and non-aneurysmal SCADs.

Aneurysmal and non-aneurysmal SCADs carry a similarly low risk of recurrent ischemic strokes and recurrent dissections. The majority of patients with strokes from both groups had a favorable functional outcome after 3 months.

**Author Contributions:** Conceptualization, M.A.; Investigation, V.K.S., H.S., P.R.M., S.W., C.I., P.B., S.J., K.A., U.F., J.G., R.M.U., B.G.S., C.J.H.C.M.v.L., G.J.d.B., H.C., M.R.H. and M.A.; Data curation, V.K.S. and B.G.S.; Writing—original draft, V.K.S. and M.R.H.; Writing—review & editing, V.K.S. and

M.R.H.; Visualization, V.K.S.; Supervision, M.A. All authors have read and agreed to the published version of the manuscript.

**Funding:** This research received no external funding.

**Institutional Review Board Statement:** This study was conducted in accordance with the Declaration of Helsinki and approved by the Ethics Committee of the canton of Bern, Switzerland (KEK 122/12).

**Informed Consent Statement:** Informed consent was obtained from all subjects involved in the study.

**Data Availability Statement:** The raw data of all patients included in this study can be made available upon request to the corresponding author and after clearance by the local ethics committee.

**Acknowledgments:** We are grateful to all three stroke teams from all three stroke centers who contributed to this study, and we would like to thank Corrado A. Bernasconi for his support and insights into the statistical analysis.

**Conflicts of Interest:** H.S., U.F., S.J., and B.G.S. were supported in the framework of the Special Program for University Medicine (SPUM-Grant 33CM30-124119), which is funded by the Swiss National Science Foundation (SNSF)). K.A. reports a grant from the Swiss National Science Foundation (SNSF), not directly related to the submitted work. M.R.H. reports grants from the Bangerter Foundation, the SITEM Support Funds, and the Swiss National Science Foundation (SNSF), all outside of the submitted work. All other co-authors report no disclosures directly related to this manuscript.

## Appendix A

**Table A1.** Sensitivity analysis with aneurysm at baseline only. Comparison of the characteristics of 1004 patients by aneurysm status, univariate analysis.

| | Number of Patients Included in Analysis | No Aneurysm | Aneurysm | Univariate Analysis |
|---|---|---|---|---|
| | *n* (% Missing Data) | Median (IQR)/ *n* (%) | Median (IQR)/ *n* (%) | *p*-Value |
| Age | 1004 (0.0) | 45.0 (38.0 to 52.0) | 44.5 (38.0 to 52.0) | 1 |
| Female sex | 1004 (0.0) | 399 (43.8) | 48 (51.1) | 0.405 |
| Days to diagnosis | 1004 (0.0) | 5.0 (2.0 to 12.0) | 7.0 (5.0 to 15.0) | 0.006 |
| Vessel affected | 1004 (0.0) | | | 0.567 * |
| ICA | | 589 (64.7) | 44 (46.8) | |
| VA | | 275 (30.2) | 28 (29.8) | |
| ICA + VA | | 46 (5.1) | 22 (23.4) | |
| Multiple dissections | 1004 (0.0) | 146 (16.0) | 34 (36.2) | 0.195 * |
| Fever and infection | 919 (8.5) | 86 (10.3) | 7 (8.0) | 0.872 |
| Minor trauma | 1001 (0.3) | 206 (22.7) | 21 (22.8) | 1 |
| Connective tissue disease | 998 (0.6) | | | 0.429 |
| Family history | | Family history | 16 (1.8) | |
| Diagnosed | | Diagnosed | 17 (1.9) | |
| Family history of ischemic stroke | 987 (1.7) | 126 (14.1) | 11 (11.8) | 0.884 |
| Smoking | 996 (0.8) | | | 0.088 |
| Former | | Former | 123 (13.6) | |
| Current | | Current | 263 (29.1) | |
| Arterial hypertension | 1003 (0.1) | 199 (21.9) | 25 (26.6) | 0.564 |
| Hypercholesterolemia | 842 (16.1) | 417 (54.2) | 31 (43.1) | 0.301 |
| Migraine | 971 (3.3) | | | 0.336 |
| Without aura | | Without aura | 167 (19.0) | |
| With aura | | With aura | 87 (9.9) | |

**Table A1.** *Cont.*

| | Number of Patients Included in Analysis | No Aneurysm | Aneurysm | Univariate Analysis |
|---|---|---|---|---|
| | *n* (% Missing Data) | Median (IQR)/ *n* (%) | Median (IQR)/ *n* (%) | *p*-Value |
| Diabetes mellitus | 1003 (0.1) | 20 (2.2) | 2 (2.1) | 1 |
| Hormonal anticonception | 431 (3.6) | 120 (31.1) | 9 (20.0) | 0.336 |
| Headache | 1002 (0.2) | 656 (72.2) | 64 (68.1) | 0.669 |
| VAS headache | 445 (38.2) | 7.0 (5.0 to 8.0) | 7.0 (6.0 to 10.0) | 0.311 |
| Thunderclap headache | 893 (11.1) | 51 (6.3) | 9 (11.0) | 0.311 |
| Cervical pain | 1001 (0.3) | 412 (45.4) | 48 (51.6) | 0.484 |
| VAS cervical pain | 248 (46.1) | 6.0 (5.0 to 8.0) | 7.0 (5.8 to 8.0) | 0.274 |
| Radiculopathy | 996 (0.8) | 5 (0.6) | 0 (0.0) | 1 |
| Spinal infarction | 1004 (0.0) | 1 (0.1) | 0 (0.0) | 1 |
| Subarachnoid hemorrhage | 1002 (0.2) | 13 (1.4) | 7 (7.4) | 0.018 |
| Horner's syndrome | 701 (0.0) | 306 (48.2) | 27 (40.9) | 0.311 |
| Tinnitus | 700 (0.1) | 62 (9.8) | 11 (16.7) | 0.311 |
| Cranial nerve palsy | 701 (0.0) | 45 (7.1) | 12 (18.2) | 0.195 |
| Amaurosis fugax | 1002 (0.2) | 69 (7.6) | 2 (2.1) | 0.284 |
| Retinal infarction | 1004 (0.0) | 5 (0.5) | 2 (2.1) | 0.311 |
| Transient ischemic attack | 1002 (0.2) | 264 (29.1) | 20 (21.3) | 0.311 |
| Number of transient ischemic attacks | 275 (3.2) | 1.0 (1.0 to 2.0) | 1.0 (1.0 to 1.5) | 0.952 |
| Ischemic stroke | 1004 (0.0) | 483 (53.1) | 26 (27.7) | <0.0001 |
| NIHSS baseline | 500 (1.8) | 5.0 (2.0 to 14.0) | 4.0 (1.0 to 7.0) | 0.417 |
| Early recurrent stroke | 946 (5.8) | 8 (0.9) | 4 (4.5) | 0.126 |
| Late recurrent stroke | 946 (5.8) | 8 (0.9) | 0 (0.0) | 1 |
| Early recurrent TIA | 947 (5.7) | 14 (1.6) | 4 (4.5) | 0.284 |
| Late recurrent TIA | 947 (5.7) | 15 (1.7) | 0 (0.0) | 1 |
| Recurrent dissection | 949 (5.5) | 37 (4.3) | 3 (3.3) | 1 |
| Rankin Scale after 3 months | 401 (21.2) | 1.0 (1.0 to 3.0) | 0.0 (0.0 to 1.0) | 0.018 |
| Death after 3 months | 818 (18.5) | 9 (1.2) | 0 (0.0) | 1 |

\* *p*-value calculated per dissection, not per patient. ICA: internal carotid artery; VA: vertebral artery; IQR: inter-quartile range.

**Table A2.** Sensitivity analysis with aneurysm at baseline only. Comparison of the characteristics of 1004 patients by aneurysm status; multivariate analysis.

| | No Dissecting Aneurysm | Dissecting Aneurysm | Multivariate Analysis | |
|---|---|---|---|---|
| | *n* (%) | *n* (%) | Odds Ratio (95% CI) | *p*-Value |
| Ischemic stroke | 483 (53.1) | 26 (27.7) | 0.36 (0.22–0.57) | <0.0001 |

CI: confidence interval.

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
