# Peer review of "Comparison of Aneurysmal and Non-Aneurysmal Spontaneous Cervical Artery Dissections in a Large Multicenter Cohort"

_ctn, doi:10.3390/ctn8020018_

Round 1

Reviewer 1 Report

Comments and Suggestions for Authors

This study investigated the natural history of dissecting cervical artery aneurysms. Some issues existed.

1.     The language needs to be improved because of some grammar issues and difficult language.

2.     Title: The current title is too vague or general and should be more specific to let the readers know what you are going to do. The title should correspond to your study purpose.

3.     The purpose at the end of the introduction should be consistent with the title and the conclusion. Why do you want to study the natural history? Please state clearly here in the introduction section.

4.     IN the discussion, “Patients with DA had significantly fewer ischemic strokes at baseline in our analysis. Although most strokes in CAD are due to embolism,20 we initially hypothesized that this finding was related to the degree of stenosis,21 postulating that protrusion in the direction of the vessel outside a DA causes less high-grade stenosis and occlusion,” what do the numbers (20 and 21) mean here? Are they citations?

5.     When you investigate the natural history, it means what happens in the study period to this disease in these patients. Has this disease remained normal or ok or deteriorated? What is the rate? It does mean that you should investigate the risk factors for the deterioration. If so, you should specify this. This is why  you should be more specific in the title. The title, the purpose and the conclusion of the study should be consistent with each other. Please check the whole structure of your article.

Comments on the Quality of English Language

revision is needed.

Author Response

  1. The language needs to be improved because of some grammar issues and difficult language.

We have thoroughly revised the manuscript and corrected grammar and spelling issues and simplified the language wherever possible.

  1. Title: The current title is too vague or general and should be more specific to let the readers know what you are going to do. The title should correspond to your study purpose.

We have changed the title to “Comparison of dissecting and non-dissecting cervical artery aneurysms in a large multicenter cohort”, trying to be as specific as possible.

  1. The purpose at the end of the introduction should be consistent with the title and the conclusion. Why do you want to study the natural history? Please state clearly here in the introduction section.

We have changed the title and adapted the introduction section according to the suggestion of the reviewer. Line 57 in the introduction further specifies the purpose of the study.

  1. IN the discussion, “Patients with DA had significantly fewer ischemic strokes at baseline in our analysis. Although most strokes in CAD are due to embolism,20 we initially hypothesized that this finding was related to the degree of stenosis,21 postulating that protrusion in the direction of the vessel outside a DA causes less high-grade stenosis and occlusion,” what do the numbers (20 and 21) mean here? Are they citations?

Thank you very much for having noted this. These are citations that were incorrectly formatted. We have corrected this and adapted the order of the references accordingly.

  1. When you investigate the natural history, it means what happens in the study period to this disease in these patients. Has this disease remained normal or ok or deteriorated? What is the rate? It does mean that you should investigate the risk factors for the deterioration. If so, you should specify this. This is why you should be more specific in the title. The title, the purpose and the conclusion of the study should be consistent with each other. Please check the whole structure of your article.

We changed the title according to the suggestion of the reviewer and focused on the comparison of dissecting and non-dissecting cervical artery aneurysms in all sections of our article.

Reviewer 2 Report

Comments and Suggestions for Authors

This is a case-control study on spontaneous dissection of the carotid and vertebral arteries in the neck, of aneurysmal and non-aneurysmal origin.

In lines 214 and 215 the numbers 20 and 21 appear, which probably refer to a number of patients with clinical conditions expressed in the text, I suggest enclosing them in parentheses.

Although the available literature on the topic is scarce in the databases, only 1 reference was written in the last 5 years. I think it should be improved a little and made more up-to-date.

In general sense there were no significant difference between the study groups, including the results and although the authors identify and record in the text their opinion that "The lower risk of ischemic strokes at baseline, which was independent of the degree of stenosis in our data, is a novel finding andmay reflect a different pathomechanisms in aneurysmal and non aneurysmal SCAD", the study does not provide significant results to science.

Author Response

  1. In lines 214 and 215 the numbers 20 and 21 appear, which probably refer to a number of patients with clinical conditions expressed in the text, I suggest enclosing them in parentheses.

Thank you very much for having noted this. These are citations that were incorrectly formatted. We have corrected this and adapted the order of the references accordingly.

  1. Although the available literature on the topic is scarce in the databases, only 1 reference was written in the last 5 years. I think it should be improved a little and made more up-to-date.

We do agree that our references include very little recent articles. However, searching for newer literature, we were unable to find any non-cited recent contribution concerning dissecting aneurysms after cervical artery dissections relevant to this article.

  1. In general sense there were no significant difference between the study groups, including the results and although the authors identify and record in the text their opinion that "The lower risk of ischemic strokes at baseline, which was independent of the degree of stenosis in our data, is a novel finding andmay reflect a different pathomechanisms in aneurysmal and non aneurysmal SCAD", the study does not provide significant results to science.

Many thanks for this comment. While it is true that the study may not have yielded groundbreaking differences between the two groups, the identification of a potentially lower risk of ischaemic strokes at baseline, independent of the degree of stenosis, and its implications for different pathomechanisms in aneurysmal and non-aneurysmal SCAD are noteworthy contributions to our understanding in this specific field. We also think that the lack of other significant differences between dissecting and non-dissecting aneurysms is a meaningful finding, because our findings may be reassuring for clinicians treating patients with extracranial dissecting aneurysms and patients suffering from extracranial dissecting aneurysms.

Reviewer 3 Report

Comments and Suggestions for Authors

First of all, I want to congratulate the authors for the immense effort behind this paper. Collecting data from three different centers is a challenging task. The paper answers a fascinating question, and the fact that this is one of the most significant cohorts makes it even more valuable. 

Regarding some recommendations:

1. In the abstract I would expand the concept of "natural history" because this can mean many things. Most people in the field would understand that you're referring to the dissection itself; however, it's always better to be as clear as possible. 

2. Based on the document, it sometimes gives more of the idea of a cross-sectional cohort rather than a case-control study. To solve this, I recommend that 1) In the methods, clarify how you defined a case (and, of course, the control). I know this seems evident in the results, but as I said before, clarity is essential.  

3. Review Table 1, number of aneurysms section for the dissecting aneurysms section. I don't think that 0 is the correct value. 

4. In Table 2. I think you should review the consistency in the labeling; some of the elements are in capital letters (Ex. Ischemic), while the rest are in lower caps. 

5.  Might be good if you separate the multivariate analysis from the rest

Author Response

  1. In the abstract I would expand the concept of "natural history" because this can mean many things. Most people in the field would understand that you're referring to the dissection itself; however, it's always better to be as clear as possible.

We did change the title and adapted lines 17 and 18 in the abstract, providing a more specific description than “natural history”.

  1. Based on the document, it sometimes gives more of the idea of a cross-sectional cohort rather than a case-control study. To solve this, I recommend that 1) In the methods, clarify how you defined a case (and, of course, the control). I know this seems evident in the results, but as I said before, clarity is essential. 

We improved the description of the analysis in the methods (mostly lines 66 to 75) to represent more clearly our definition of cases and controls.  

  1. Review Table 1, number of aneurysms section for the dissecting aneurysms section. I don't think that 0 is the correct value.

We do think the value is formally correct. In Table 1, under dissecting aneurysms, 0 patients have 0 aneurysms, because otherwise they would not be grouped in the dissecting aneurysm group. However, since this line is confusing to the reader and it may be unnecessary information, we did remove it.

  1. In Table 2. I think you should review the consistency in the labeling; some of the elements are in capital letters (Ex. Ischemic), while the rest are in lower caps.

We did review and correct the labeling.

  1. Might be good if you separate the multivariate analysis from the rest

We separated the multivariate analysis from the rest according to the suggestion of the reviewer.

Round 2

Reviewer 1 Report

Comments and Suggestions for Authors

I am not sure if  dissecting and non-dissecting cervical artery aneurysms are the same as  aneurysmal and non-aneurysmal SCAD. please check. 

Author Response

  1. I am not sure if dissecting and non-dissecting cervical artery aneurysms are the same as  aneurysmal and non-aneurysmal SCAD. please check.

Thank you very much for having noted this. We have corrected the title accordingly.

Reviewer 2 Report

Comments and Suggestions for Authors

Dear authors, we thank you for considering our timely suggestion for correcting your article. At this time the peer review process has achived a better scientific article that will be af great interest to the scientific community.

Thank you very much.

Angel J. Lacerda MD Ph.D.

Author Response

Dear reviewer

We highly appreciate your valuable feedback, input and suggestions and thank you for taking the time to thoroughly review our manuscript.

Yours sincerely

The authors